# Consumer Perception of Food Fraud in Serbia and Montenegro

**DOI:** 10.3390/foods13010053

**Published:** 2023-12-22

**Authors:** Ilija Djekic, Nada Smigic

**Affiliations:** Department of Food Safety and Quality Management, Faculty of Agriculture, University of Belgrade, Nemanjina 6, 11080 Belgrade, Serbia; nadasmigic@agrif.bg.ac.rs

**Keywords:** fraudulent activities, types of food fraud, consumer awareness, consumer trust

## Abstract

The main objective of this study was to investigate how food fraud is perceived among consumers in Serbia and Montenegro. A total of 1264 consumers from the two countries participated in an online survey during the second half of 2022, using Google forms^®^. In the Serbian population, older or highly educated respondents are aware of different types of fraudulent activities such as substitution, mislabeling, concealment, and counterfeiting. Dilution is mostly recognized by women, the younger population, and students. Consumers believe that trust is the most important factor when purchasing food. The highest level of agreement regarding food fraud is that such activities may pose serious health risks to consumers, and that food inspection services are the most responsible actors in the food chain continuum. When it comes to purchasing food, open green markets are most trustworthy, followed by hypermarkets. Concerning the types of food, fish is most susceptible to fraud, followed by olive oil. This study builds upon existing knowledge of food consumers about food fraud in Europe.

## 1. Introduction

Although there are several definitions of food fraud outlined by different authors and sources such as Spink and Moyer [1], international standards [2,3] or legislation [4], food fraud is generally considered as an economically motivated fraudulent behavior that affects the product in one or more ways. Théolier et al. [5] state that all fraudulent definitions are supported by four pillars: (i) legal breaches; (ii) misleading information; (iii) intentionality; and (iv) financial benefits. Fraudulent mechanisms include dilution, substitution, concealment, unapproved enhancement, mislabeling, grey market production and counterfeiting [6]. However, a study performed by Spink et al. [7] outlines that even among experts, some terms (like food adulteration) raise confusion and there is a lack of agreement on a common definition. When it comes to food law, food safety is the focus worldwide, with more than one law outlining all these requirements, while only minimal quality requirements provide the framework to ensure that food is not subject to any fraudulent practices [8]. Robson et al. [9] emphasize that there is lack of legally accepted terminology regarding food fraud in the European Union, leading to a certain discrepancy in academia and among different regulatory bodies.

When it comes to “figures and numbers”, the databases most commonly used are the Rapid Alert System for Food and Feed (RASFF) in the EU, used as open access platforms, and the Economically Motivated Adulteration hosted in the USA, requiring subscription [10]. These types of databases are mainly populated with results from different control/inspection activities. The main intention of such information sources is to provide added value to data analysis and shift food fraud activities from detection and correction to prevention [11]. In parallel, such information helps in informing consumers how they can be cheated by fraudsters [10]. Since food fraud is evolving in terms of types and financial benefits, some authors categorize it as intentional food crime [12,13].

### Literature Review and Reseach Background

To analyze the association between consumer awareness and knowledge and food fraud, the authors of this study performed a text-mining research on the available literature using the VOSViewer tool for bibliometric analysis [14]. A set of metadata have been captured from scientific publications indexed in the scientific database Web of Science. The search was refined by document type, including only research and review papers. Two combinations of keywords were used: (i) “food fraud” + “knowledge” and (ii) “food fraud” + “awareness”. Other similar terms such as economically motivated adulteration or fraudulent activities have not been taken into account for this text-mining research. The search yielded 99 manuscripts published from 2010 to the present time. The cut-off criterion for including keywords in network visualization was that it occurred at least 10 times.

Figure 1 displays a network visualization based on the titles, abstracts, and keywords of selected articles. It shows three clusters presented in different colors. The red cluster mainly deals with food authenticity and the geographic origin of food. The blue cluster outlines food safety-based tools used to combat food fraud such as traceability and vulnerability assessments. Finally, the green cluster is related to consumers and their behavior, preferences, and perceptions.

When it comes to a deeper analysis of the selected publications, only a limited number of papers had focused on consumers. Moreira et al. [15] analyzed consumer knowledge about food fraud and food labels in Portugal. Another study was performed in Canada studying the level of awareness and trust among consumers regarding fraudulent and counterfeit food [16]. Both studies point to age and education as the main demographic factors influencing both knowledge and awareness. However, awareness and knowledge about food fraud should be understood differently, awareness as “overall understanding,” while knowledge as “thorough understanding” [5].

Another research dimension was using the “halo effect” on olive oil scandals and studying the effect of prior knowledge about food fraud and consumer behavior [17]. Théolier et al. [5] in their review paper clearly emphasized that consumer knowledge about food fraud is limited, subject to various psychosocial influences, and biased. In parallel, the link between consumer perceptions and food fraud opens new dimensions of consumer risk literature [18]. 

The geographic distribution of published papers from the text mining search shows that the research focus was on countries of the European Union, USA and China. Only one paper covering this region was found, namely Serbia and Croatia, analyzing food fraud perception of food companies [19]. Different cultures have different influences on food choices [20], which are affected by the traditions and social habits of food consumers [21]. In countries like Serbia and Montenegro that have a similar cultural dimension [22,23], food choices are strongly influenced by socially accepted values [24] where high level of health concerns affect food choices [25]. Bearing in mind that Serbia and Montenegro are not members of the EU (Serbia with a population of 7.1 million, Montenegro with a population of 0.6 million [26]), the main objective of this paper was to build upon existing knowledge of food consumers about food fraud and reveal how fraud is perceived among consumers in these two countries of southeast Europe. This research had three specific goals that were deployed from the main objective: (i) to study the awareness of the consumers on food fraud; (ii) to analyze their trust in food purchasing places; and (iii) to reveal types of food most prevalent to fraud. 

## 2. Materials and Methods

### 2.1. Field Survey

The data used in this research were collected from an organized online survey in Serbia and Montenegro using Google forms^®^ during the second half of the year 2022. This type of surveys enabled parallel consumer research in multiple countries due to its speed and potential to reach a larger number of respondents [27]. As this study employed convenience sampling involving participants accessible and available online, and since it was open to anyone to participate, it can be classified as an ‘unrestricted self-selected survey’ giving the possibility to approached individuals to choose whether to participate [28]. 

Respondents were recruited mainly from networks of family contacts and professional connections on social media followed by further dissemination of the questionnaire online. A total of 1273 respondents initially participated in the survey, but only 1264 completed questionnaires were further processed (969 respondents from Serbia and 295 from Montenegro). Considering the population of the two countries, with a defined confidence level of 95%, and margin of error of 3.1% for Serbia and 5.6% for Montenegro, the sampling size is adequate [29,30]. Demographic characteristics of the sample in the two countries are shown in Table 1.

### 2.2. Questionnaire 

The questionnaire consisted of the following sections: (i) demographic characteristics; (ii) single and multiple responses to questions on how well consumers are informed about the various types of fraudulent activities, media coverage of international fraud incidents, the type of food products that were subject of fraud and the reasons for it, and perceptions of food fraud in their countries; (iii) statements about food fraud using a 5-point Likert scale (1—totally disagree, 2—disagree, 3—neutral, 4—agree, 5—totally agree); (iv) places where people are most/least confident about fraud; and (v) food most/least affected by fraud. The questionnaire was carefully developed by integrating findings from various reports and articles on the subject [16,19,31,32]. The original questions were modified to adequately reflect the local market scenario and included specific examples relevant to local food products and purchasing behavior. By adapting the questions to relate to local food products, popular supermarket chains, and well-known open markets, we aimed to promote a deeper understanding of food fraud in the context of people’s preferences and consumption habits. This approach increases the relevance of the questionnaire and allows for a more accurate assessment of consumer awareness of food authenticity and potential fraudulent practices in the region.

### 2.3. Data Processing and Statistical Methods

Multiple response questions and demographic characteristics (country, gender, age) were subject to correspondence analyses or the chi-square test for association to test the homogeneity of the classes. The first correspondence analysis shows that the first two dimensions explained 79.5% of the variance in the original contingency data (Figure 2). The second analysis reveals that the first two dimensions explain 93.1% of the variance in the original contingency data (Figure 3).

Likert scale data from the questionnaires were processed using non-parametric statistical tests. A two-step cluster analysis was employed to categorize the statements deployed by various demographic parameters (country, gender, age, education) as categorical variables. The Mann–Whitney U test was used to understand whether statements reveal statistically significant differences between the clusters.

Most/least trust in food shopping places and most/least types of food prevalent to fraud were analyzed using the Best–Worst score by calculating the difference of Total(Most)–Total(Least) and dividing by the number of respondents [33,34]. In parallel, “Most–Least percentages” (percentage of attributes chosen as “most”, percentage of attributes chosen as “least”, and percentage of attributes not chosen as “most”/“least”) were also captured [35]. Both Best–Worst analyses were considered as ranks (“most” was assigned “3”, “least” was assigned “1” and if not selected was assigned “2”) and were subject to Friedman’s analysis followed by a comparison test for rank sums employing the Wilcoxon signed-rank test. 

Statistical significance was set at *p* < 0.05. Statistical software used were SPSS Statistics 23 and Minitab 17.

## 3. Results and Discussion

### 3.1. Demography of the Sample

Table 1 displays the demographic characteristics of the sample by country of origin, gender, age, household members, education, and place of residence. Female respondents (60.7%) prevailed compared to male respondents (38.0%) and those who chose not to provide information (1.3%). The age distribution showed that two thirds of the respondents were below 40 years old. Most of the respondents live in 4-person households. Two thirds of the respondents were students and highly educated respondents. The majority (81.0%) live in urban areas.

### 3.2. Awareness about Food Fraud

The first multiple response question was “What food fraud have you heard of?”, giving the respondents the opportunity to select from among seven potentially fraudulent activities. The results of the survey showed that almost half of respondents were familiar with terms such as food fraud (57.3%), substitution (55.1%), dilution (52.5%), and mislabeling (47.9%) when asked about various potentially fraudulent activities in the food industry. However, our results showed that respondents were less aware of the other forms of fraudulent practices mentioned in the survey These results highlight a clear discrepancy in the level of awareness of different types of food fraud among respondents, with some specific fraudulent activities being better known than others. As displayed in Figure 2, the main difference between fraudulent activities was confirmed by dilution as opposed to grey market and enhancement (component 1) and fraud as opposed to substitution, concealment, mislabeling and counterfeit (component 2). Respondents from Montenegro and with lower education had opposite attitudes compared to respondents from Serbia and respondents with higher education. The younger population (below 40) had opposed attitudes compared to the older population. Consumers from Montenegro with lower education only recognized fraud in general as the predominant fraudulent activity. Dilution was mostly associated with women, the younger population, and students. In the Serbian population, older respondents and those with a high level of education linked fraudulent activities with substitution, mislabeling, concealment and counterfeit. Although there is a certain degree of misunderstanding of different types of fraud activities, consumers in general do understand what food fraud is [5]. Also, the more they are aware of food practices, the more they are aware of food fraud risks [36]. 

The respondents were asked to identify fraud incidents they are aware of. Almost half of the respondents stated that they were not aware of any of the listed cases of food fraud when asked to select one of the listed incidents (47.8%). Among those who have heard of some food fraud incidents, the most cited was the 2013 case of illegal adulteration of horsemeat in the UK with 29.1% [37], followed by the 2009 outbreak of peanut butter contaminated with Salmonella in the USA with 24.4% [38], the deliberate addition of melamine to diluted raw milk to increase protein content in China in 2008 with 24.2% [39], and the outbreak of rapeseed oil denatured with aniline in Spain in 1981 with 17.9% [40]. It is interesting that the 1981 Spanish incident caused the highest number of mortality and morbidity in Europe’s food fraud history [18], but was not recognized by many respondents. Since this incident occurred several decades ago, before most respondents were born or at an age where they could follow news and historical events, it is plausible that the incident was not as particularly covered or discussed in their recent memory. A survey in the UK just after the horsemeat outbreak showed three main outcomes: (i) lower level of purchasing products with processed meat; (ii) lower level of confidence regarding the entire meat chain; and (iii) need for more strict control and improved traceability [41]. This highlights the importance of understanding that consumers perceive correlation between the complexity of global food chains and fraud events [42], but also point to the role of food inspection services. 

The chi square test revealed that there was no statistically significant association between multiple responses to known food fraud incidents and gender (χ^2^ = 10.427, *p* > 0.05), country (χ^2^ = 5.095, *p* > 0.05), and education (χ^2^ = 9.870, *p* > 0.05), but there was a statistically significant association between multiple responses to known food fraud incidents and the age of respondents (χ^2^ = 31.018, *p* < 0.05). As mentioned above, the possible explanation could be related to the fact that most respondents were under 40 years old and were unaware of the incidents that had occurred in the past. This corresponds with the findings of a Canadian survey that older consumers are to a certain degree more sensitive to food fraud due to their life experiences [16]. 

The most important mechanism to avoid buying fraudulent food is trusting the food producer (43.7% of responses), followed by reading the labels (15.7%). This is in line with the study of Kendall et al. [43] that stated there is a clear connection between food purchasing patronage and trust. In parallel, labelling plays an important role in consumers’ attempt to verify claims such as organic production or country of origin and to improve traceability [44]. Misleading food labels pave the way for dissatisfaction and lack of trust and/or confidence in both the product and the producer [18]. 

However, trust of consumers related to existing food controls is disturbed due to various fraudulent events that occurred in the previous years [5]. Almost two thirds of respondents (61.9%) confirmed that they know they have bought fake food, while 30.1% believe they have probably bought fraud food). A lack of knowledge paves the way for consumers to play a game of hide and seek with other parts of the food chain, leading to unexpected purchasing patronage [5]. When participants in our survey were asked to name previously purchased fraudulent products, the majority named meat (42.5%) and dairy (35.3%), followed by honey (22.9%), and fish (24.4%). These results are consistent with global data showing that these food categories are particularly susceptible to fraud. Participants’ beliefs were primarily based on personal suspicion (58.9%), own research (18.7%), media reports (10.5%) or recall notices from manufacturers (7.0%) and retailers (4.8%). The correspondence analysis (Figure 3) shows that students and young people identify different types of food subject to fraud than highly educated and older respondents. Types of food subject to fraud were categorized into a group consisting of animal origin food and fruits and vegetables (mainly women), by a population below 40 years of age, (mainly from Serbia), as opposed to wine and honey (older and highly educated) and olive oil (Montenegro). 

When asked about the reasons for product fraud, half of the respondents identified substitution with cheaper raw materials (25.0%) and dilution (21.8%). Mislabeling (incorrect indication that the food is organic—17.1%, incomplete list of ingredients—10.8% and insufficient indication of country of origin—10.6%) was also emphasized by respondents. The chi square test showed that there was a statistically significant association between these multiple response set of answers and all demographic categories: country (χ^2^ = 161.078, *p* < 0.05), gender (χ^2^ = 27.515, *p* < 0.05), age (χ^2^ = 46.565, *p* < 0.05), and education of respondents (χ^2^ = 52.318, *p* < 0.05). This confirms that various demographic groups associated different types of fraud (Figure 3). When it comes to statistics of food fraud events, this type of data are uncertain as food fraud is hardly detected by consumers and retail may also be the victim of fraudulent activity that occurs earlier in the supply chain [45]. Some studies identify mislabeling as the main fraud threat [12,46]. 

### 3.3. Statements about Food Fraud

The two-step cluster analysis with country, gender, age, and education as categorical variables resulted in two clusters (Table 2). The overall results showed that the highest level of agreement was in relation to fraudulent activities that may pose serious health risks to consumers (4.3) and that respondents consider food inspection services as the most responsible actors in the food chain (4.1), followed by food processors (4.0). According to Van Rijswijk and Frewer [47], when consumers are subject to fraud, one third will stop buying such products but only 10% will contact any food inspection authority. Van Ruth et al. [48] clearly identify control measures as guardianship for the entire supply chain. It is also worth mentioning that respondents believe that there is more fraudulent food on the market than five years ago (4.0). This perception could be due to increased awareness due to media coverage, growing consumer awareness, or an increase in reported cases influencing their perception of an increased presence of fraudulent products on the market. However, this perception alone does not necessarily confirm an actual increase in food fraud, but rather indicates a change in consumer sentiment and awareness of the issue over time.

Cluster 1 (536 respondents) consists of the older population (above 40 years old) and those who live in Montenegro. In general, this cluster expresses a higher agreement with all statements compared to cluster 2 (728 respondents) which consists of younger respondents and mainly people from Serbia. A statistically significant difference was found between two clusters for all statements (*p* < 0.05). Although the two countries have been part of the same country in the past [49], and are of similar cultural background [22], it is considered that Montenegrins are slightly more conservative than Serbians. Therefore, similar levels of agreements between the Montenegrins and older citizens may be expected.

The highest level of agreement in Cluster 1 was obtained for the statement that fraudulent activities may cause serious health problems for consumers (4.8), concern about fraudulent food imported from China (4.7), and the responsibility of food inspection services (4.6). Some studies reveal that fraud food imported to the European Union originated from China [50]. Due to the complexity of global supply chains and the fact that governments cannot achieve full control of imports, the food industry relies on second and third party audits [51]. The Global Food Safety Initiative recognizes all major food safety standards (FSSC 22000, BRC and IFS) and released its guidance documents to assess vulnerability to food fraud and define mitigation strategies to combat food fraud [6,52]. However, existing food safety management systems are proactive in addressing process/product nonconformities [53] but have limited tools to combat food fraud presuming honesty in the supply chain [48]. The highest scoring statements in cluster 2 relate to health issues and food fraud (3.9), the role of food inspection services (3.8), and food producers (3.7). Although food fraud is mainly driven by economic motives, mislabeling is one of the most often fraudulent activities that can pose a serious health risk [54]. 

### 3.4. Best–Worst Analysis

Table 3 shows the respondents’ subjective priority for the most and least trusted purchasing places. Open green markets were identified as the most trustful places for purchasing, followed by hypermarkets. Both purchasing places were chosen as the most trusted places in more than 25% of the cases. On the contrary, online shopping and fast food were identified as the least trusted.

The significance of the ‘S’ score is that it has the potential to identify the most trusted purchasing locations, as rated by the respondents. It emphasizes the relative power of the purchasing place. Positive values mean that the purchasing place was selected as the most trusted more often than as the least trusted, in contrast to negative values where the least trusted selection prevailed. The values of zero indicates that the number of the most and least trusted location is equal [55].

The visual representation of the ‘S’ score shows that five places were positively rated with a statistically significant difference in the perceived priority of the most and least trusted purchasing places, χ^2^ = 1271.185, *p* < 0.05. The results also showed that the ranking of open green markets as the most trusted compared to hypermarkets and specialized shops was statistically significant. Local shops, retail, and restaurants were ranked similarly, although with slightly different scores. There were statistically significant differences for the least trusted purchasing places—fast food and online shopping. The ‘S’ score for both clusters followed the same pattern. Some studies show that in urban areas, the most preferable and trustful purchasing places are supermarkets, as opposed to rural areas where fresh markets prevail [56]. This relationship of trust is mainly established by means of ‘reciprocity’ in short supply chain with direct connection between the customer and the supplier or producer [18]. One interesting phenomenon is that of the consequential actions that occur upon a breach of trust when consumers seek not only for alternative purchasing places, but also for different types of supply chains [51]. 

Table 4 shows how respondents rate the types of food that are the most/least susceptible to fraud. Fish is rated as the most susceptible to fraud in more than 25% of cases, followed by olive oil. This corresponds with an industrial survey in Serbia and Croatia identifying olive oil and spices with the highest fraud risks [19]. According to these results, cereals, sauces, and fillings are the least fraud-prone foods. From a scientific point of view, olive oil was mostly covered in the literature through different fraudulent dimensions [57], followed by seafood, concurring with our results. However, organic food (regardless of the type of food), is considered as most fraudulent food item. It is interesting that some studies confirm a connection between trust and knowledge about potential fraudulent risks associated with different types of food [18]. When it comes to the physical state of the products, powder and liquid products are more vulnerable opposed to solid food [19].

The visual representation of the ‘S’ score shows that seven types of food are positively rated, with a statistically significant difference between the most and least fraud-prone foods, χ^2^ = 753.253, *p* < 0.05. Further analysis (Wilcoxon signed-rank test) showed that there was a statistically significant difference between four food groups: (i) fish; (ii) olive oil; (iii) meat and meat products, honey, milk and dairy products, fruit juices and organic food; and (iv) wine, spices, coffee and tea, cereals and sauces and fillings. The ‘S’ score for both clusters followed the same pattern for the first two types of food (‘i’ and ‘ii’), with a slight change of the order of the remaining two groups of food (‘iii’ and ‘iv’).

## 4. Conclusions

This study clearly shows that age and education play an important role in how food fraud is perceived by food consumers. Older and more educated respondents are more aware of different types of fraudulent activities and perceive fraud as a serious risk, as opposed to the young population that has a more relaxed approach. This calls for developing a clear strategy for informing and raising awareness among the young population.

Consumers identify food fraud as an activity with the potential to cause health risks. Food inspection services are identified as being more responsible for preventing food fraud than food producers. Therefore, it is of utmost importance to further develop legislation protecting the food supply chain continuum. Open green markets are considered as the most trustworthy places for food purchasing, followed by hypermarkets clearly connecting both short (local) and long (global) supply chains. When it comes to different types of food, fish and olive oil are recognized as the most susceptible to fraud.

This study provides useful information to different stakeholders in the food chain continuum regarding food fraud. For the food sales sector, it provides insights regarding trust when purchasing food. For consumers it enlightens the need for media coverage of various food fraud incidents with the aim of helping them understand different food crime strategies. Finally, it can provide aid to food policy makers on employing more effective prevention measures in combating food fraud. As such, these results build on the evolving literature on food fraud in Europe, providing useful information about this issue in southeast Europe.

Future research should focus on the aftermath of exposed food fraud. The first limitation of the study may be the fact that female populations slightly prevailed. However, as female populations are the main food customers, the authors believe that this limitation did not jeopardize the results in any way. The second limitation of the study is that it was a form of convenience sampling.

## Figures and Tables

**Figure 1 foods-13-00053-f001:**
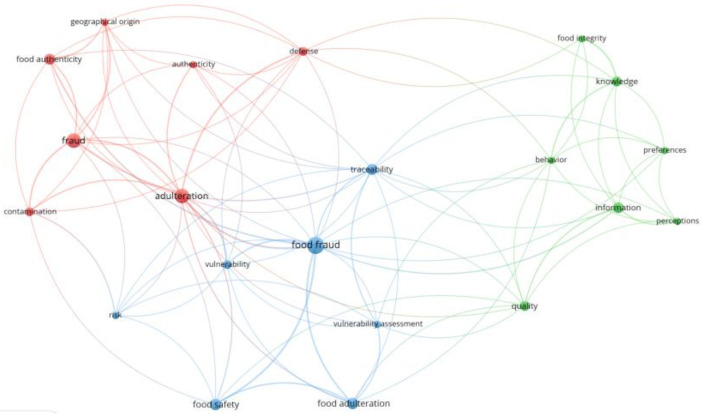
Network visualization of the correlation between food fraud and knowledge/awareness.

**Figure 2 foods-13-00053-f002:**
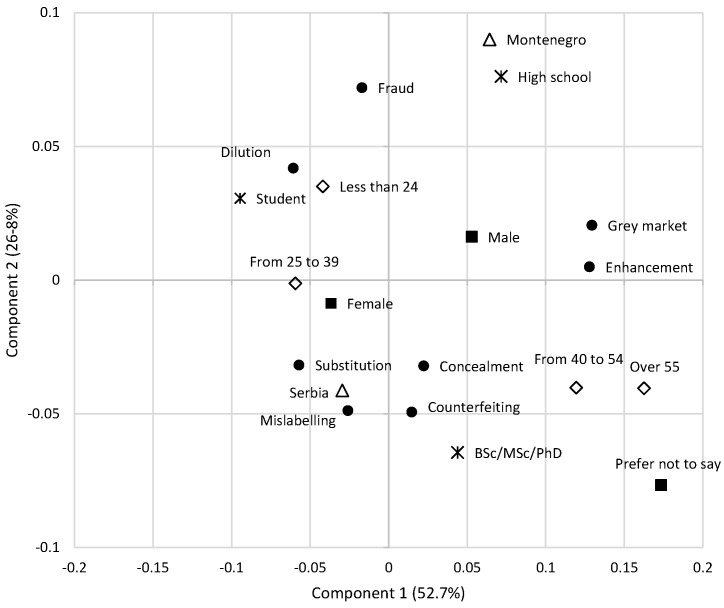
Multiple responses to the question: “What food fraud have you heard of?”. The biplot displays the results of the correspondence analysis based on the data collected from 1264 respondents. Legend of symbols: ● fraudulent activities; △ country; ■ gender; ◇ age; 
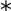
 education.

**Figure 3 foods-13-00053-f003:**
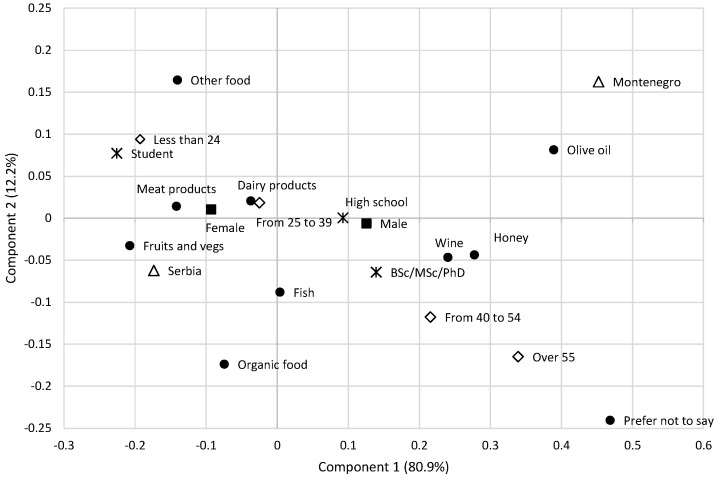
Multiple responses to “which type of food you purchased was subject to fraud?” The biplot displays the results of the correspondence analysis based on the data collected from 1264 respondents. Legend of symbols: ● fraud food; △ country; ■ gender; ◇ age; 
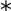
 education.

**Table 1 foods-13-00053-t001:** Demography per country (N = 1264).

	SRB (*n* = 969)	MNE (*n* = 295)	Total N (%)
Gender			
Male	328 (33.8%)	152 (51.5%)	480 (38.0%)
Female	627 (64.7%)	140 (47.5%)	767 (60.7%)
Prefer not to say	14 (1.4%)	3 (1.0%)	17 (1.3%)
Age			
Below 24 years	379 (39.1%)	82 (27.8%)	461 (36.5%)
25–39 years	360 (37.2%)	124 (42.0%)	484 (38.3%)
40–54 years	172 (17.8%)	66 (22.4%)	238 (18.8%)
Above 55 years	58 (6.0%)	23 (7.8%)	81 (6.4%)
Household members			
1 person	43 (4.4%)	6 (2.0%)	49 (3.9%)
2 persons	95 (9.8%)	31 (10.5%)	126 (10.0%)
3 persons	190 (19.6%)	58 (19.7%)	248 (19.6%)
4 persons	412 (42.5%)	125 (42.4%)	537 (42.5%)
More than 4	229 (23.6%)	75 (25.4%)	304 (24.1%)
Education			
High school graduets	201 (20.7%)	80 (27.1%)	281 (22.2%)
Undergraduete students	398 (41.1%)	87 (29.5%)	485 (38.4%)
Bachelor’s degree holders and beyond	370 (38.2%)	128 (43.4%)	498 (39.4%)
Residence			
Urban	777 (80.2%)	247 (83.7%)	1024 (81.0%)
Rural	87 (9.0%)	33 (11.2%)	120 (9.5%)
Suburbia	105 (10.8%)	15 (5.1%)	120 (9.5%)

Legend: *n* represents the number of interviewees; (%) represents their share in the sample. Country codes: Serbia—SRB; Montenegro—MNE.

**Table 2 foods-13-00053-t002:** Description of the two clusters in terms of country, gender, age and education (N = 1264).

Respondents Characteristics		Cluster 1 (*n* = 536)	Cluster 2 (*n* = 728)	Total (N = 1264)
Country	Serbia	366 (37.8%)	603 (62.2%)	969 (100%)
	Montenegro	170 (57.6%)	125 (42.4%)	295 (100%)
Gender	Male	219 (45.6%)	261 (54.4%)	480 (100%)
	Female	311 (40.5%)	456 (59.5%)	767 (100%)
	Prefer not to say	6 (35.3%)	11 (64.7%)	17 (100%)
Age	Below 24 years of age	171 (37.1%)	290 (62.9%)	461 (100%)
	Between 25 and 39 years of age	191 (39.5%)	293 (60.5%)	484 (100%)
	Between 40 and 54 years of age	131 (55%)	107 (45%)	238 (100%)
	Over 55 years of age	43 (53.1%)	38 (46.9%)	81 (100%)
Education	High school and lower	133 (47.3%)	148 (52.7%)	281 (100%)
	Student	167 (34.4%)	318 (65.6%)	485 (100%)
	Bachelor and higher	236 (47.4%)	262 (52.6%)	498 (100%)
Food safety statements		Mean ± StD ^1^	Mean ± StD ^1^	Mean ± StD ^1^|Mode ^2^
I am concerned about food fraud and the sale of such products in the domestic market.		4.5 ± 0.7 ^a^	3.3 ± 1.0 ^b^	3.8 ± 1.1|5.0
I believe that some fraudulent activities can cause serious health problems for consumers.		4.8 ± 0.5 ^a^	3.9 ± 1.1 ^b^	4.3 ± 1.0|5.0
I am concerned about fraudulent food produced in my country.		4.4 ± 0.7 ^a^	3.1 ± 1.0 ^b^	3.7 ± 1.1|5.0
I am concerned about fraudulent food imported from neighbouring countries.		4.5 ± 0.7 ^a^	3.2 ± 1.0 ^b^	3.7 ± 1.1|5.0
I am concerned about fraudulent food imported from the European Union.		4.4 ± 0.8 ^a^	3.1 ± 1.1 ^b^	3.7 ± 1.2|5.0
I am concerned about fraudulent food imported from China.		4.7 ± 0.6 ^a^	3.4 ± 1.1 ^b^	4.0 ± 1.1|5.0
I recognize the food inspection services as the most responsible for preventing food fraud.		4.6 ± 0.8 ^a^	3.8 ± 1.1 ^b^	4.1 ± 1.1|5.0
I recognize food producers being primarly responsible for preventing food fraud.		4.4 ± 0.9 ^a^	3.7 ± 1.1 ^b^	4.0 ± 1.1|5.0
I recognize consumers as the most responsible for preventing food fraud.		3.2 ± 1.4 ^a^	2.8 ± 1.2 ^b^	2.9 ± 1.3|3.0
When I buy food, I am aware of the risk of buying fraudulent food.		4.2 ± 0.9 ^a^	3.4 ± 1.1 ^b^	3.7 ± 1.1|5.0
I believe that there is more fraudulent food on the market than five years ago.		4.5 ± 0.8 ^a^	3.5 ± 1.1 ^b^	4.0 ± 1.1|5.0
I believe that food produced by multinational companies is the least affected by food fraud.		3.7 ± 1.3 ^a^	3.2 ± 1.1 ^b^	3.4 ± 1.2|3.0

The Mean values ± Standard deviations ^1^ and modes ^2^ were obtained from the raw data. Note: Items denoted with different letters are significantly different at the level of 5%. Likert scale: (1) “Strongly disagree”, (2) “Disagree”, (3) “No opinion”, (4) “Agree”, (5) “Strongly agree”.

**Table 3 foods-13-00053-t003:** Subjective priority of most and least trustful purchasing places: Best–Worst scaling report-frequency counts and standardized average score considering the entire sample (N = 1264) and two clusters.

**Shopping Places**	**Number of “Most Trustful”**	**Number of “Least Trustful”**	**Distribution [%]**			**Cluster 1 ‘S’ Score (*n* = 536)**	**Cluster 2 ‘S’Score** **(*n* = 728)**	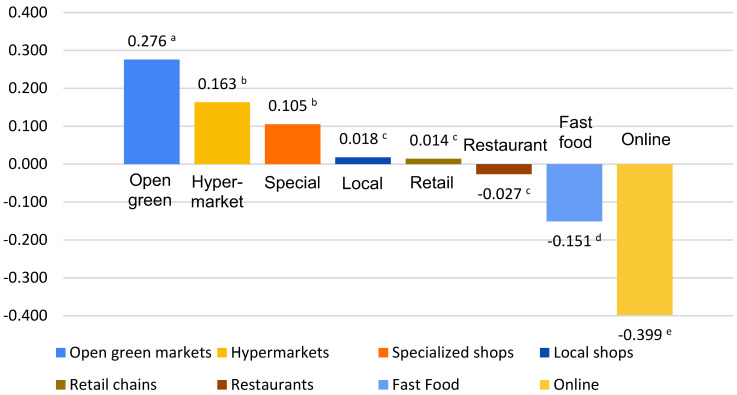
			**Most Trustful**	**Least Trustful**	**Not Chosen**			
Open green markets	400	51	31.65	4.03	64.32	0.250	0.295	
Hypermarkets	357	151	28.24	11.95	59.81	0.183	0.148	
Specialized shops	164	31	12.97	2.45	84.57	0.146	0.076	
Local shops	77	54	6.09	4.27	89.64	0.022	0.015	
Retail chains	62	44	4.91	3.48	91.61	0.022	0.008	
Restaurants	63	97	4.98	7.67	87.34	−0.028	−0.026	
Fast Food	66	257	5.22	20.33	74.45	−0.168	−0.139	
Online	75	579	5.93	45.81	48.26	−0.427	−0.378	

Legend: Values marked with different letter are statistically different (*p* < 0.05).

**Table 4 foods-13-00053-t004:** Subjective priority of most and least food prevalent to fraud: Best–Worst scaling report-frequency counts and standardized average score considering the entire sample (N = 1264) and two clusters.

**Type of Food**	**Number of “Most Prevalent”**	**Number of “Least Prevalent”**	**Distribution [%]**			**Cluster 1 ‘S’ Score (*n* = 536)**	**Cluster 2 ‘S’ Score** **(*n* = 728)**	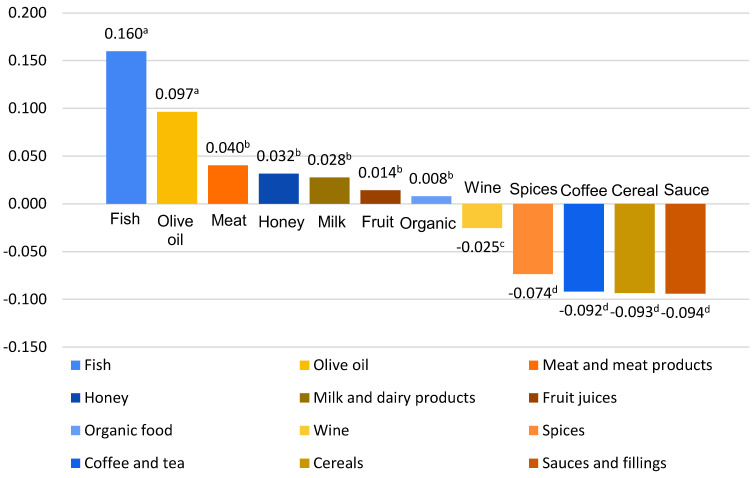
			**Most Prevalent**	**Least Prevalent**	**Not Chosen**			
Fish	357	155	28.24	12.26	59.49	0.063	0.231	
Olive oil	209	87	16.53	6.88	76.58	0.112	0.085	
Meat and meat products	123	72	9.73	5.70	84.57	0.080	0.011	
Honey	178	138	14.08	10.92	75.00	0.037	0.027	
Milk and dairy products	123	88	9.73	6.96	83.31	0.076	−0.008	
Fruit juices	55	37	4.35	2.93	92.72	0.022	0.008	
Organic food	80	70	6.33	5.54	88.13	0.039	−0.015	
Wine	19	51	1.50	4.03	94.46	−0.024	−0.026	
Spices	11	104	0.87	8.23	90.90	−0.095	−0.058	
Coffee and tea	28	144	2.22	11.39	86.39	−0.119	−0.071	
Cereals	11	129	0.87	10.21	88.92	−0.108	−0.082	
Sauces and fillings	70	189	5.54	14.95	79.51	−0.084	−0.102	

Values marked with different letter are statistically different (*p* < 0.05).

## Data Availability

Data is contained within the article.

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
