# Peer review of "Consumer Perception of Food Fraud in Serbia and Montenegro"

_foods, 2023, doi:10.3390/foods13010053_

Round 1

Reviewer 1 Report

Comments and Suggestions for Authors

The MS reports the results of analyzing the reply given by 1264 consumers about a questionnaire on Consumer perception of food fraud in Serbia and Montenegro. The topic is worth interest and the sample size used for the analysis is adequate and sufficient. The MS is also well supported by a robust statistical analysis background but there are several sections to clarify before to consider the MS for publication.

The introduction is confusing and does not report the exhaustive background of information to "justify" the need to undergo the study in the two selected countries. Preliminary studies carried on in the two countries or surroundings should be reported in the introduction section as well as the lack of enough information that could be the reason for carrying on the study itself. For example, in the reference list and in the MS text it is cited a paper about a study carried out in Serbia and Croatia but the study is not reported in the introduction.  In addition, the figure 1 reported in the introduction is unclear and it is not explained if it refer to the study itself (in this case should be moved to results and an explanation of the methods should be given in the material and methods) or if it was taken from another source (in this case should be cited). See the pdf attached for more detailed comments.

The material and method section is too confusing and much information and details about the analyses reported in the results are missing. On the other way, the results of some of the analyses cited in the material and method are not clearly reported in the results. These two sections should be revised considering all the analyses step by step. See the pdf attached for more detailed comments. 

Results are in general explicative but some additional analyses could better clarify and support the findings. For example repeat also the correspondence analysis for the two country samples separated could be useful to understand whether the effect of age is evident in both countries or if the results are biased by the higher number of respondents from Serbia than Montenegro. Another concern is about the demography category ''students'' used for "age". I have some doubts about the category "student" because it does not give a real explanation of the education level of the respondents. they can be students at University (first class) and then have a high school education but also be a master student and then have a bachelor's education. It is not clear how these cases and the category "students" were considered. Some more details are requested. See the pdf attached for more detailed comments.

There are two Figure 1 in the text. I strongly suggest (again) to delete the one reported in the introduction if not related to the study. See the pdf attached for more detailed comments. 

For all these reasons, in my opinion, the MS should be reconsidered after major revision. 

Author Response

Reviewer #01

The MS reports the results of analyzing the reply given by 1264 consumers about a questionnaire on Consumer perception of food fraud in Serbia and Montenegro. The topic is worth interest and the sample size used for the analysis is adequate and sufficient. The MS is also well supported by a robust statistical analysis background but there are several sections to clarify before to consider the MS for publication.

Thank you. We have provided explanation for all the comments (and the pdf file)

The introduction is confusing and does not report the exhaustive background of information to "justify" the need to undergo the study in the two selected countries. Preliminary studies carried on in the two countries or surroundings should be reported in the introduction section as well as the lack of enough information that could be the reason for carrying on the study itself. For example, in the reference list and in the MS text it is cited a paper about a study carried out in Serbia and Croatia, but the study is not reported in the introduction.  In addition, the figure 1 reported in the introduction is unclear and it is not explained if it refers to the study itself (in this case should be moved to results and an explanation of the methods should be given in the material and methods) or if it was taken from another source (in this case should be cited). See the pdf attached for more detailed comments.

In line with your comments, and comments from other reviewers, we have revised the Introduction section. Within the Introduction section, we have defined a subsection “Literature review and research background”. Figure 1 is populated by the authors of this study, and it refers to this study and our bibliometric research employing VOSViewer.

The material and method section is too confusing and much information and details about the analyses reported in the results are missing. On the other way, the results of some of the analyses cited in the material and method are not clearly reported in the results. These two sections should be revised considering all the analyses step by step. See the pdf attached for more detailed comments. 

In line with your comments, and comments from other reviewers, we have revised this section.

Results are in general explicative but some additional analyses could better clarify and support the findings.

We have added more explanations in this section.

For example repeat also the correspondence analysis for the two country samples separated could be useful to understand whether the effect of age is evident in both countries or if the results are biased by the higher number of respondents from Serbia than Montenegro.

It’s not statistically viable to perform a correspondence analysis separately. However, we provided some additional explanation in the Results and discussion regarding the two countries and age of the sample.

Another concern is about the demography category ''students'' used for "age". I have some doubts about the category "student" because it does not give a real explanation of the education level of the respondents. they can be students at University (first class) and then have a high school education but also be a master student and then have a bachelor's education. It is not clear how these cases and the category "students" were considered. Some more details are requested. See the pdf attached for more detailed comments.

Thank you for the comment. We have corrected groups according to education, into "High school graduates": persons who have successfully completed high school. "Undergraduate students" means students who are currently enrolled in an undergraduate degree programme and are pursuing a bachelor's degree. "Bachelor's degree graduates and beyond": respondents who have completed a bachelor's degree or have higher educational qualifications, such as a master's degree, doctorate or professional certification.

There are two Figure 1 in the text. I strongly suggest (again) to delete the one reported in the introduction if not related to the study. See the pdf attached for more detailed comments. 

We have revised the numbering of the figures as per your comment, and comment from another reviewer. The Figure for bibliometric analysis is created by ourselves and it relates to the study, so we have left it. 

For all these reasons, in my opinion, the MS should be reconsidered after major revision. 

In line with your comments (and additional pdf), as well as the comments from other reviewers, we hope we have improved the paper.

Reviewer 2 Report

Comments and Suggestions for Authors

 Abstract

Line 8. Please make: In the Serbian ………

Line 9. Counterfeiting

Line 10.  …… the younger population

Line 14. trustworthy ……..

Line 15. …… the types of food

Please include the methods used in the abstract. Please also give a conclusion for your study in the abstract.

 Introduction

Line 22. Théolier et al. [4] state ……..

Line 26. ….. ‘Interesting’…… Please cancel.

Line 28. ………. safety is in the focus worldwide…… please cancel ‘in’

Line 33. Please correct this way: ………..performed text-mining research on the literature using VOSViewer tool………

Line 40. …. in- between….. please cancel ‘in-‘

Line 42-47. Please acknowledge and cite for others work.

Line 48. …… comes to a deeper analysis of the publications, a limited number of papers…. Please correct this way.

Line 52. Both studies point to age and education as the main….. Please correct this way.

Line 54. …… should be understood differently…. Please correct this way.

Line 56. Another research dimension was using the….. Please correct this way.

Line 58. …… clearly emphasized that ….. Please correct this way.

Line 61-65. Please rewrite the object of the study and make it concise and brief.

 Materials and Methods

Line 70. This type of surveys became very popular for parallel consumer research in multiple countries due to its speed and potential to reach a larger number of respondents [13].

In the scientific paper there is no need to promote, and better more to cite the work. Please rewrite or can cancel line 70-71.

Line 103. Please insert the formula used in the Friedman Analysis.

Line 104. Please write the type of SPSS and Minitab versions used.

A critical revision of the methods by including some physicochemical properties of the products sounds for the readership of Foods Journal.

 Results and discussion

Line 113. Two thirds of the respondents were students and highly educated respondents. It would be nice if you included the level and specialty of the respondents. Are they from food science and nutrition? Are they taking a sensory evaluation course? Mentioning the specialization of the respondents is important. Do you feel that some of the judgments undertaken are measured by navies without having orientation or training?

Line 114. The majority (81.0%) live in urban areas. It's better to cancel cities.

Line 123-137. After reading this paragraph, readers might raise the question, So what? Accordingly, based on the stated paragraph, the authors should give a one- or two-sentence remark at the end.

Line 142-154. Please state your result 1st then support your work with the other reports. Please also check for grammar.

Line 159. Please check this sentence to see if it goes with the paragraph: 'This is probably because most of the respondents were under 40 years old and were not aware of the incident in Spain in 1981'.

Line 176. Please state the question 1st 

Line 176-183. Please check for grammar.

Line 201-210. Please check the grammar and tenses.

Line 231. It would be nice if you merged Tables 1 and 2, as they are similar in respondent characteristics. And separately handle Food safety statements.

Line 273. Please make it: According to these results, cereals, sauces and fillings are the least fraud-prone foods.

Line 275. Please make it: From a scientific point of view, olive oil was mostly covered in literature through different fraudulent dimensions.

Line 279. Please make it. ….. knowledge about potential fraudulent risks associated with different types of food……

 Conclusion

Line 290-294. Please correct the verbs accordingly. Older and more educated respondents are more aware of different types of fraudulent activities and perceive fraud as a serious risk. Opposed to a young population that has a more relaxed approach. This calls for developing a clear strategy for informing and raising awareness among the young population.

Line 290-294. Don’t you think this is a vague idea that the awareness of the older and educated respondents to fraudulent is more that is coming from exposure, experience, and academics? Is this researchable? I think this is the universal truth.

Line 295. Please correct this way……with the potential to cause health risks.

Line 296. Please correct this way………. identified as being more responsible for preventing food fraud than food producers.

Line 299. Please correct this way: Open green markets are the most trustworthy places ……

Line 301. Please correct this way ……. recognized as the most susceptible to fraud.

Comments on the Quality of English Language

Extensive editing of English language required

Author Response

Reviewer #02

Abstract

Line 8. Please make: In the Serbian ………

Thank you. This has been corrected.

Line 9. Counterfeiting

Thank you. This has been corrected.

Line 10.  …… the younger population

Thank you. This has been corrected.

Line 14. trustworthy ……..

Thank you. This has been corrected.

Line 15. …… the types of food

Thank you. This has been corrected.

Please include the methods used in the abstract. Please also give a conclusion for your study in the abstract.

In line with your comment, and comments from other reviewers, the abstract has been revised.

Introduction

Line 22. Théolier et al. [4] state ……..

Thank you. This has been corrected.

Line 26. ….. ‘Interesting’…… Please cancel.

Thank you. This has been corrected.

Line 28. ………. safety is in the focus worldwide…… please cancel ‘in’

Thank you. This has been corrected.

Line 33. Please correct this way: ………..performed text-mining research on the literature using VOSViewer tool………

Thank you. This has been corrected.

Line 40. …. in- between….. please cancel ‘in-‘

Thank you. This has been corrected.

Line 42-47. Please acknowledge and cite for others work.

We have added additional references in the Introduction section

Line 48. …… comes to a deeper analysis of the publications, a limited number of papers…. Please correct this way.

Thank you. This has been corrected.

Line 52. Both studies point to age and education as the main….. Please correct this way.

Thank you. This has been corrected.

Line 54. …… should be understood differently…. Please correct this way.

Thank you. This has been corrected.

Line 56. Another research dimension was using the….. Please correct this way.

Thank you. This has been corrected.

Line 58. …… clearly emphasized that ….. Please correct this way.

Thank you. This has been corrected.

Line 61-65. Please rewrite the object of the study and make it concise and brief.

As some other reviewwers also commented the objectives, we have revised it to covers all comments.

Materials and Methods

Line 70. This type of surveys became very popular for parallel consumer research in multiple countries due to its speed and potential to reach a larger number of respondents [13]. In the scientific paper there is no need to promote, and better more to cite the work. Please rewrite or can cancel line 70-71.

This has been rephrased. Thank you.

Line 103. Please insert the formula used in the Friedman Analysis.

Since we employed several statistical methods (chi-square, Friedman Analysis, cluster analysis, Mann-Whitney U test, Wilcoxon signed-rank test), we have additionally clarified the statistical part (in line with comments from several reviewers). We believe providing formulas for all the methods would be confusing for readers.

Line 104. Please write the type of SPSS and Minitab versions used.

Thank you. This has been corrected.

A critical revision of the methods by including some physicochemical properties of the products sounds for the readership of Foods Journal.

Thank you for this comment. However, as the questionnaire didn’t explore any physicochemical properties of the products, it was hard to provide this type of information since we didn’t ask the respondents. However, we have commented issue (physical state of the products), after Table 4.

Results and discussion

Line 113. Two thirds of the respondents were students and highly educated respondents. It would be nice if you included the level and specialty of the respondents. Are they from food science and nutrition? Are they taking a sensory evaluation course? Mentioning the specialization of the respondents is important. Do you feel that some of the judgments undertaken are measured by navies without having orientation or training?

We have provided explanation that this study employed convenience sampling involving participants accessible and available online, and we classified it as 'unrestricted self-selected survey'. We have emphasized as a limitation that our study was a form of convenience sampling

Line 114. The majority (81.0%) live in urban areas. It's better to cancel cities.

Thank you. This has been corrected.

Line 123-137. After reading this paragraph, readers might raise the question, So what? Accordingly, based on the stated paragraph, the authors should give a one- or two-sentence remark at the end.

We have provided additional explanation in this section.

Line 142-154. Please state your result 1st then support your work with the other reports. Please also check for grammar.

Thank you. This has been corrected.

Line 159. Please check this sentence to see if it goes with the paragraph: 'This is probably because most of the respondents were under 40 years old and were not aware of the incident in Spain in 1981'.

Thank you. This has been corrected.

Line 176. Please state the question 1st 

Thank you. This has been corrected.

Line 176-183. Please check for grammar.

Thank you, it has been corrected.

Line 201-210. Please check the grammar and tenses.

Thank you, it has been corrected.

Line 231. It would be nice if you merged Tables 1 and 2, as they are similar in respondent characteristics. And separately handle Food safety statements.

We understand your concern. However, to keep the cluster with the statements, we have left the tables as they were.

Line 273. Please make it: According to these results, cereals, sauces and fillings are the least fraud-prone foods.

Thank you. This has been corrected.

Line 275. Please make it: From a scientific point of view, olive oil was mostly covered in literature through different fraudulent dimensions.

Thank you. This has been corrected.

Line 279. Please make it. ….. knowledge about potential fraudulent risks associated with different types of food……

Thank you. This has been corrected.

Conclusion

Line 290-294. Please correct the verbs accordingly. Older and more educated respondents are more aware of different types of fraudulent activities and perceive fraud as a serious risk. Opposed to a young population that has a more relaxed approach. This calls for developing a clear strategy for informing and raising awareness among the young population.

Thank you. This has been corrected.

Line 290-294. Don’t you think this is a vague idea that the awareness of the older and educated respondents to fraudulent is more that is coming from exposure, experience, and academics? Is this researchable? I think this is the universal truth.

Despite the almost universal acceptance of this term, we would like to emphasize this aspect in order to clarify why the younger population may not perceive crises such as that of 1981.

Line 295. Please correct this way……with the potential to cause health risks.

Thank you. This has been corrected.

Line 296. Please correct this way………. identified as being more responsible for preventing food fraud than food producers.

Thank you. This has been corrected.

Line 299. Please correct this way: Open green markets are the most trustworthy places ……

Thank you. This has been corrected.

Line 301. Please correct this way ……. recognized as the most susceptible to fraud.

Thank you. This has been corrected.

Reviewer 3 Report

Comments and Suggestions for Authors

Dear authors,

thank you for the opportunity to review your interesting paper on food fraud. Please find below a few of my comments, which are hopefully helpful.

Introduction and Literature review: 

Line 18-65: Can the authors please revisit the introduction and consider the following elements.  Please provide more country context towards – Serbia and Montenegro. Why is the food fraud discussion particular relevant for this SE-European context. Are there many incidences of food fraud or other food related illegal practices. Please back up with stats.  Can the authors explain a bit the food-consumer culture of these countries that food fraud and its consequences are put in perspective

Line 32-46: Delete here. Put in the method or results section. Diturbs the text flow currently.

Line 48-60: Shift into the literature review section and present the state of the art . This is completely missing

Maybe helpful for your work:

Kendall, H., Clark, B., Rhymer, C., Kuznesof, S., Hajslova, J., Tomaniova, M., ... & Frewer, L. (2019). A systematic review of consumer perceptions of food fraud and authenticity: A European perspective. Trends in Food Science & Technology94, 79-90.

Visciano, P., & Schirone, M. (2021). Food frauds: Global incidents and misleading situations. Trends in Food Science & Technology114, 424-442.

Manning, L. (2016). Food fraud: Policy and food chain. Current Opinion in Food Science10, 16-21.

Manning, L., & Soon, J. M. (2016). Food safety, food fraud, and food defense: a fast evolving literature. Journal of food science81(4), R823-R834.

Robson, K., Dean, M., Haughey, S., & Elliott, C. (2021). A comprehensive review of food fraud terminologies and food fraud mitigation guides. Food Control120, 107516.

Material and Methods:

Line 73-77: Can the authors explain and justify their sampling approach. This is convience sampling? Or is it purposive sampling and the criteria are missing. Please elaborate

Line 79-87: Can the authors be more precise with their item and scale development. Please indicate which scales and items stem from the two indicated studies. What adjustments were made. What is the authors own work?

Data analysis: You have a good sample, why are not executing a multivariate analysis. For the best-worst analysis, maybe consider 

Louviere, J. J., Flynn, T. N., & Marley, A. A. J. (2015). Best-worst scaling: Theory, methods and applications. Cambridge University Press.

Bir, C., Delgado, M., & Widmar, N. (2022). Development, Implementation, and Evaluation of a More Efficient Method of Best-Worst Scaling Data Collection. Agricultural and Resource Economics Review51(1), 178-201.

Results and Discussion:

The entire section needs improvement as the results are not put in perspective with the recent body of literature. A discussion is in essence missing

Conclusion:

Please incoperate best practice recommandations for practitioners and consumers.

Please critically relect on your own work and acknowledge limitation--> Sampling and analysis

Please indicate the theoretical contributions of your work. What is the novelty and merit of this work

Author Response

Reviewer #03

Dear authors,

Thank you for the opportunity to review your interesting paper on food fraud. Please find below a few of my comments, which are hopefully helpful.

Introduction and Literature review: 

Line 18-65: Can the authors please revisit the introduction and consider the following elements.  Please provide more country context towards – Serbia and Montenegro. Why is the food fraud discussion particular relevant for this SE-European context. Are there many incidences of food fraud or other food related illegal practices. Please back up with stats. 

We have provided rational for selecting Serbia and Montenegro. In parallel, we have provided stats about the two countries.

Can the authors explain a bit the food-consumer culture of these countries that food fraud and its consequences are put in perspective.

Thank you for this comment. We have provided additional explanation regarding food-consumer culture in the section “Literature review and research background”.

Line 32-46: Delete here. Put in the method or results section. Disturbs the text flow currently.

We have created a subsection “Literature review and research background” as per comment of other reviewers, and revised the introduction part.

Line 48-60: Shift into the literature review section and present the state of the art. This is completely missing.

We have created a subsection “Literature review and research background” as per comment of other reviewers, and revised the introduction part.

Maybe helpful for your work:

Kendall, H., Clark, B., Rhymer, C., Kuznesof, S., Hajslova, J., Tomaniova, M., ... & Frewer, L. (2019). A systematic review of consumer perceptions of food fraud and authenticity: A European perspective. Trends in Food Science & Technology94, 79-90.

Visciano, P., & Schirone, M. (2021). Food frauds: Global incidents and misleading situations. Trends in Food Science & Technology114, 424-442.

Manning, L. (2016). Food fraud: Policy and food chain. Current Opinion in Food Science10, 16-21.

Manning, L., & Soon, J. M. (2016). Food safety, food fraud, and food defense: a fast evolving literature. Journal of food science81(4), R823-R834.

Robson, K., Dean, M., Haughey, S., & Elliott, C. (2021). A comprehensive review of food fraud terminologies and food fraud mitigation guides. Food Control120, 107516.

Thank you for these references. They were helpful. We have included them to improve the introduction section.

Material and Methods:

Line 73-77: Can the authors explain and justify their sampling approach. This is convience sampling? Or is it purposive sampling and the criteria are missing. Please elaborate.

We have provided explanation regarding sample size and sampling approach. Thank you for this comment.

Line 79-87: Can the authors be more precise with their item and scale development. Please indicate which scales and items stem from the two indicated studies. What adjustments were made. What is the authors own work?

We have provided additional explanations regarding the development of the questionnaire.

Data analysis: You have a good sample, why are not executing a multivariate analysis. For the best-worst analysis, maybe consider. 

Louviere, J. J., Flynn, T. N., & Marley, A. A. J. (2015). Best-worst scaling: Theory, methods and applications. Cambridge University Press.

Bir, C., Delgado, M., & Widmar, N. (2022). Development, Implementation, and Evaluation of a More Efficient Method of Best-Worst Scaling Data Collection. Agricultural and Resource Economics Review51(1), 178-201.

Thank you for this comment. We are aware of different methods for processing best-worst analysis. However, we believe the statistical processing we performed is appropriate and provides useful conclusions, in line with the guides from our statistical department.

Results and Discussion:

The entire section needs improvement as the results are not put in perspective with the recent body of literature. A discussion is in essence missing.

We have added more discussion in this section.

Conclusion:

Please incorporate best practice recommendations for practitioners and consumers.

We have provided a paragraph about the implications of this study to different stakeholders in the food chain.

Please critically reflect on your own work and acknowledge limitation--> Sampling and analysis

Thank you. We have provided information about the sampling method (form of convenience sampling) as a limitation in the Conclusion. 

Please indicate the theoretical contributions of your work. What is the novelty and merit of this work.

We provided a sentence on the contribution or our work to the evolving literature on food fraud in Europe.

Reviewer 4 Report

Comments and Suggestions for Authors

I recommend the manuscript. The authors have conducted a comprehensive online survey on consumer perceptions of food fraud, specifically in 'Serbia' and 'Montenegro.' The analysis, based on responses from 1,264 consumers, delves into their psychology regarding food adulteration and emphasizes the importance of raising awareness. The outcomes presented in this report are anticipated to significantly benefit researchers engaged in similar studies. Furthermore, it is commendable that the manuscript has undergone meticulous proofreading, with only a few minor errors identified.

1.      The numbering of figures needs correction, with Figure 1 repeated twice.

2.      Figures 2 and 3 appear a bit confusing. I would suggest that the authors simplify the notation for better comprehension.

3.      In Tables 3 and 4, the color scheme of the bar plot may be confusing. I recommend naming the respective bars at the bottom for clarity.

4.      In Tables 3 and 4, since the author has already mentioned that columns 4, 5, and 6 represent percentage values, there may be no need to repeat this information each time.

Author Response

Reviewer #04

I recommend the manuscript. The authors have conducted a comprehensive online survey on consumer perceptions of food fraud, specifically in 'Serbia' and 'Montenegro.' The analysis, based on responses from 1,264 consumers, delves into their psychology regarding food adulteration and emphasizes the importance of raising awareness. The outcomes presented in this report are anticipated to significantly benefit researchers engaged in similar studies.

Thank you.

Furthermore, it is commendable that the manuscript has undergone meticulous proofreading, with only a few minor errors identified:

  1. The numbering of figures needs correction, with Figure 1 repeated twice.

This has been corrected. Thank you.

  1. Figures 2 and 3 appear a bit confusing. I would suggest that the authors simplify the notation for better comprehension.

As we had several factors, we had to use many symbols (we understand it may be a bit confusing). Therefore, we have provided additional explanations in the text regarding the two tables.

  1. In Tables 3 and 4, the color scheme of the bar plot may be confusing. I recommend naming the respective bars at the bottom for clarity.

This has been corrected. Thank you.

  1. In Tables 3 and 4, since the author has already mentioned that columns 4, 5, and 6 represent percentage values, there may be no need to repeat this information each time.

This has been corrected. Thank you.

Reviewer 5 Report

Comments and Suggestions for Authors

Thank you for the opportunity to review the manuscript entitled “Consumer perception of food fraud in Serbia and Montenegro”, submitted for possible publication to Foods. The manuscript deals with an interesting topic in the field of food consumption, specifically food fraud. 

Abstract. The abstract is quite incomplete. It does not report the research context and the purpose of the research, it starts in medias res with the data collection and the main insights of the research, but is requires several additional information related to the context, the aims and scope, the methods adopted, etc. Last, to whom is the research addressed?

Introduction. The section “Introduction” lacks a clear and comprehensive description of the research context and of the literature background of the topic, as well as updated statistics and facts related to food frauds at the global and the local scale. Specifically, statistics and facts related to the issue in Serbia and Montenegro are crucial to justify the ratio behind this research and to highlight the novelty/originality of the research. 

LL. 32-37. The adoption of the text-mining research and the VOSviewer tool for bibliometric analysis in the section “Introduction” is quite unappropriated. Several information should be provided in order to describe how the bibliometric search has been performed, namely: inclusion and exclusion criteria, keywords combination, journals, timeline, etc. The simple statement provided ad LL. 34-38 is quite insufficient to develop the research context in the field. I would move this analysis to a specific section entitled “Literature review and research background”. 

LL. 61-65. The description of the purpose of the research, in the light of the development of the previous theoretical context (LL. 42-61) is not consistent and requires additional improvements. In addition, it is not clear why the authors intend to investigate Serbia and Montenegro. 

Materials and methods. The research relies on which sampling strategy? Please, highlight it as LL. 73-75 and also justify such a choice, by considering previous studies on the topic and also including advantages/disadvantages of such a sampling.

LL. 79-87. The questionnaire development requires additional references and details. Specifically, could please the authors define how the variables investigated at point (ii) have been developed, as well as their specific references? It should be adopted a more scientific (and rigorous) approach in describing the questionnaire development. On the other hands, it is difficult to comprehend the scientific basis for the questions asked to the participants. Furthermore, what about the statement about food fraud (LL. 83-84)? Which kind of statement? Last, in the field of the data processing and statistical method (LL. 89-105), if the authors are willing to, could they include a figure, which summarized the data analysis process? 

Results and discussion. L. 120 identifies the question “What food fraud have your heard of?” and soon after the authors introduce some possible answers, namely “seven potentially fraudulent activities”. Could you please describe (and discuss) them? 

In addition, the research lacks a clear description that identifies similarities and differences between Serbian and Montenegrins. In its absence, it is not clear to understand similarities and differences in their answers (and perception).  Should these two groups be treated as one group, or there are significant differences which justify different perceptions among them?

In the field of discussion, the authors should include some managerial and public authorities’ implications, as to understand how to address consumers’ perception toward a reduction of food frauds. In synthesis, how this research is useful to readers, practitioners, public authorities, etc.?

Limitations should be enlarged, also in the field of the statistical methods adopted to analyze data and on the sampling strategy, which I suppose is a convenience one.

Author Response

Reviewer #05

Thank you for the opportunity to review the manuscript entitled “Consumer perception of food fraud in Serbia and Montenegro”, submitted for possible publication to Foods. The manuscript deals with an interesting topic in the field of food consumption, specifically food fraud. 

Thank you.

Abstract. The abstract is quite incomplete. It does not report the research context and the purpose of the research, it starts in medias res with the data collection and the main insights of the research, but is requires several additional information related to the context, the aims and scope, the methods adopted, etc. Last, to whom is the research addressed?

In line with your comments, and comments from other reviewers, we have revised the Abstract.

Introduction. The section “Introduction” lacks a clear and comprehensive description of the research context and of the literature background of the topic, as well as updated statistics and facts related to food frauds at the global and the local scale. Specifically, statistics and facts related to the issue in Serbia and Montenegro are crucial to justify the ratio behind this research and to highlight the novelty/originality of the research. 

In line with your comments, and comments from other reviewers, we have expanded the introduction section. In parallel, we have provided information about Serbia and Montenegro, and why we performed the study in these two countries.

  1. 32-37. The adoption of the text-mining research and the VOSviewer tool for bibliometric analysis in the section “Introduction” is quite unappropriated. Several information should be provided in order to describe how the bibliometric search has been performed, namely: inclusion and exclusion criteria, keywords combination, journals, timeline, etc. The simple statement provided ad LL. 34-38 is quite insufficient to develop the research context in the field. I would move this analysis to a specific section entitled “Literature review and research background”. 

Thank you for this comment. We have created a new subsection as you proposed, and provided additional information.

  1. 61-65. The description of the purpose of the research, in the light of the development of the previous theoretical context (LL. 42-61) is not consistent and requires additional improvements. In addition, it is not clear why the authors intend to investigate Serbia and Montenegro. 

In line with your comments, and comments from other reviewers, we have explained why we performed the study in these two countries.

Materials and methods. The research relies on which sampling strategy? Please, highlight it as LL. 73-75 and also justify such a choice, by considering previous studies on the topic and also including advantages/disadvantages of such a sampling.

Thank you for this comment. In line with your comment, and comments from other reviewers, we have provided additional information about the sampling method.

  1. 79-87. The questionnaire development requires additional references and details. Specifically, could please the authors define how the variables investigated at point (ii) have been developed, as well as their specific references? It should be adopted a more scientific (and rigorous) approach in describing the questionnaire development. On the other hands, it is difficult to comprehend the scientific basis for the questions asked to the participants. Furthermore, what about the statement about food fraud (LL. 83-84)? Which kind of statement?

We have provided additional explanations regarding the development of the questionnaire.

Last, in the field of the data processing and statistical method (LL. 89-105), if the authors are willing to, could they include a figure, which summarized the data analysis process? 

As there were several comments on the statistical processing from different reviewers, we have tried to provide explanation in the text (to satisfy all reviewers). In parallel, as there are 3 figures ad 2 figures in Tables, we believe another figure would not provide added value but rather confuse the readers.  

Results and discussion. L. 120 identifies the question “What food fraud have your heard of?” and soon after the authors introduce some possible answers, namely “seven potentially fraudulent activities”. Could you please describe (and discuss) them? 

We have added more explanation in this section.

In addition, the research lacks a clear description that identifies similarities and differences between Serbian and Montenegrins. In its absence, it is not clear to understand similarities and differences in their answers (and perception).  Should these two groups be treated as one group, or there are significant differences which justify different perceptions among them?

Within the cluster analysis we have provided additional information how to understand the similarities / differences of the two neighboring countries.

In the field of discussion, the authors should include some managerial and public authorities’ implications, as to understand how to address consumers’ perception toward a reduction of food frauds. In synthesis, how this research is useful to readers, practitioners, public authorities, etc.?

In the conclusion section, we have provided a paragraph on the practical implications of this study to different stakeholders.

Limitations should be enlarged, also in the field of the statistical methods adopted to analyze data and on the sampling strategy, which I suppose is a convenience one.

Thank you. We have provided this limitation in the Conclusion.

Round 2

Reviewer 1 Report

Comments and Suggestions for Authors

The authors have revised the MS following the given comments and suggestions. The current version of the MS is clearer and more explicative and, in my opinion, can be accepted for publication in this form

Reviewer 3 Report

Comments and Suggestions for Authors

My comments were all considered. The manuscript has been improved. I have no further suggestions. Best of luck

Reviewer 5 Report

Comments and Suggestions for Authors

Thank you for the opportunity to review the revised version of the manuscript entitled "Consumer perception of food fraud in Serbia and Montenegro". The authors have addressed the reviewer's comments and suggestions to improve the manuscript. The authors have provided additional details related to the context in the section "Introduction" and have enhanced the section "Materials and methods", especially for the description of the questionnaire. In addition, the authors have revised the section "Results" highlighting the originality of the research and the limitations. In its current form, I can suggest the acceptance for publication of the manuscript into Foods.